# Domain Generalization for Domain-Linked Classes

## Abstract

Domain generalization (DG) focuses on transferring domain-invariant knowledge from multiple source domains (available at train time) to an *a priori* unseen target domain(s). This task implicitly assumes that a class of interest is expressed in multiple source domains (*domain-shared*), which helps break the spurious correlations between domain and class and enables domain-invariant learning. However, in real-world applications, classes may often be expressed only in a specific domain (*domain-linked*), which leads to extremely poor generalization performance for these classes. In this work, we introduce this task to the community and develop an algorithm to learn generalizable representations for these domain-linked classes by transferring useful representations from domain-shared classes. Specifically, we propose a **F**air and c**ON**trastive feature-space regularization algorithm for **D**omain-linked DG, `FOND`. Rigorous and reproducible experiments with baselines across popular DG tasks demonstrate our method and its variants' ability to accomplish state-of-the-art DG results for domain-linked classes, given sufficient number of domain-shared classes. Complementary to these contributions, we develop theoretical insights for this task and practical insights for domain-linked class generalizability in real-world settings.

## 1 Introduction

*Domain generalization* (DG) aims to learn discriminative representations that can generalize to data distributions (domains) different from those observed during training, i.e. *out-of-distribution*. Specifically, here the target domain is assumed to be *unseen* during training. Given this goal, the guiding principle in modern DG algorithms is to learn representations that are invariant to source domains, and hence generalizable to unseen targets (Ye et al., 2021). As a result, recent works aim to explicitly reduce the representation discrepancy between multiple source-domains (Zhou et al., 2023), by leveraging distribution-alignment (Rame et al., 2022; Nguyen et al., 2021), domain-discriminative adversarial networks (Kim et al., 2023; Zhang et al., 2021a), domain-based feature-alignment (Wang et al., 2023; Ruan et al., 2022; Kim et al., 2021), and meta-learning and few-shot approaches (Qin et al., 2023; Gu et al., 2022; Shu et al., 2021; Zhang et al., 2021b; Li et al., 2018a).

Existing DG methods explicitly rely on classes being observed in multiple source-domains and/or focus only on the overall accuracy. In the real-world however, classes of interest may often be observed in a specific domain (*domain-linked*, $\mathcal{Y}_L$), setting it apart from those observed in multiple domains (*domain-shared*, $\mathcal{Y}_S$); see Fig. 1a. These lead to generalization challenges in applications including, healthcare (Chen et al., 2021), autonomous driving (Piva et al., 2023), and fraud detection (Ataabadi et al., 2022), where classes/anomalies of interest may *only* have been observed in particular demographics, regions etc., resulting in large performance discrepancies between $\mathcal{Y}_L$ and $\mathcal{Y}_S$.

Since domain-linked classes are only observed in one domain, models which aim to utilize domain-linked data often encounter spurious correlations between the domain and the class (Lynch et al., 2023; Zhang et al., 2022a). Thus the learned representations exhibit extreme bias towards domain-specific features. This challenge is only exacerbated in the DG setting where we aim to learn representations that can generalize to an unseen target domain. This results in poor performance on these classes in practice; see for instance the domain-linked versus domain-shared performance discrepancy in Fig. 1b. Consequently, we seek to specifically improve the generalizability of domain-linked classes. This task, to the best of our knowledge has not been studied in the literature.

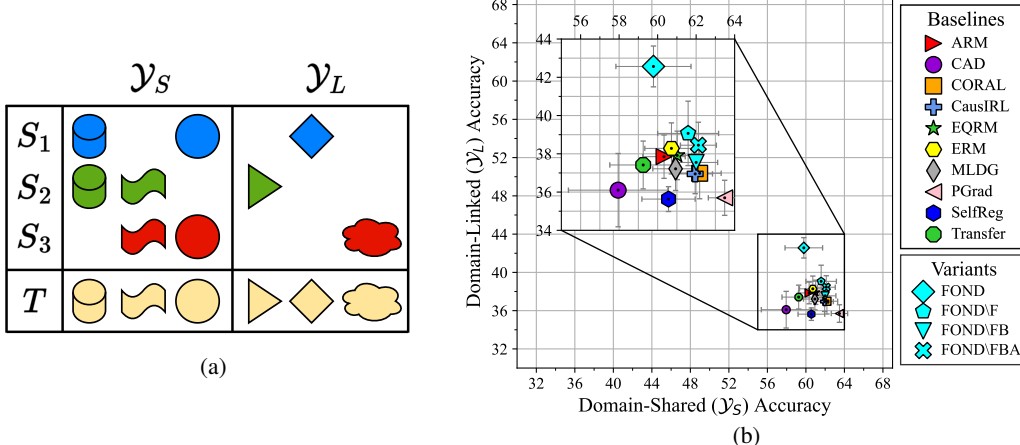

Figure 1: **Illustrating domain-linked ($\mathcal{Y}_L$) and domain-shared ($\mathcal{Y}_S$) classes and resulting performance discrepancies.** Panel (a) illustrates a shape classification task with domain-linked and domain-shared classes; the domains are represented by color. During training some classes are expressed in multiple domains (e.g. circle) while others are expressed in only one domain (e.g. triangle). Panel (b) communicates the performance discrepancy between $\mathcal{Y}_L$ and $\mathcal{Y}_S$ across all datasets.

Notwithstanding these challenges, recent advances in real-world machine learning draw from the success of pretraining with classes/objectives different from downstream tasks (He et al., 2022). This begs the question – *can we transfer useful representations from domain-shared classes to domain-linked classes?* To begin, we analyze the factors impacting domain-linked class performance theoretically. We then motivate and develop a contrastive and fairness-based objective to disentangle spurious correlations between domain and class, and propose FOND, (*Fair and cONtrastive Domain-linked learning*), to answer this question in the affirmative.

Specifically, we draw insights from recent works on *fairness* (Pham et al., 2023; Makhlouf et al., 2021; Wang et al., 2020) to learn generalizable representations from $\mathcal{Y}_S$ for $\mathcal{Y}_L$ classes. Note that this is different from the goal of classic DG fairness – achieve similar outcomes across protected attributes (e.g., gender). On the contrary, we use fairness to ensure that the model can be moved away from learning domain-specific features that can lead to spurious correlations for $\mathcal{Y}_L$. Thus leveraging fairness as a way to learn generalizable representations for domain-linked classes. To complement this objective, we develop a contrastive learning objective that regularizes the pairwise relationships between same-class-inter-domain and different-class-intra-domain training samples.

We rigorously evaluate FOND via detailed ablations, and comparisons on three standard DG benchmark datasets – PACS (Li et al., 2017), VLCS (Fang et al., 2013), and, OfficeHome (Venkateswara et al., 2017) – across ten DG baselines, include the SOTA (Wang et al., 2023; Eastwood et al., 2022). We find that indeed domain-linked class performance improves with the presence of a high enough number of domain-shared classes, thus accomplishing domain-invariant representation learning. We observe that that ERM is still a strong baseline, but FOND and its variants achieve a remarkable overall performance improvement of +9.3 over ERM on an average (26.9% improvement), with a gain of 39.2% on VLCS. This demonstrates that even observing other classes in diverse domains can immensely help domain-linked classes!

## 2 RELATED WORKS

We briefly introduce domain generalization works related to this paper and identify the research gap this paper seeks to study. To reiterate, DG aims to learn a machine learning model that predicts well on distributions different from those seen during training. To achieve this, DG methods typically aim to minimize the discrepancy between multiple source-domains (Ye et al., 2021).

**Data manipulation** techniques primarily focus on data augmentation and generation techniques. Typical augmentations include affine transformations in conjunction with additive noise, cropping and so on (Shorten & Khoshgoftaar, 2019; He et al., 2016). Other methods include simulations (Tobin et al., 2017; Yue et al., 2019; Tremblay et al., 2018), gradient-based perturbations like CrossGrad (Shankar et al., 2018), adversarial augmentation (Volpi et al., 2018) and image mixing (e.g. CutMix

(Mancini et al., 2020), Mixup (Zhang et al., 2018) and Dir-mixup (Shu et al., 2021)). Furthermore, generative models using VAEs GANs are also popular techniques for diverse data generation (Anoosheh et al., 2017; Zhou et al., 2020; Somavarapu et al., 2020; Huang & Belongie, 2017). Since the model generalizability is a consequence of training data diversity (Vapnik, 2000), FOND and other approaches should be used in conjunction. It is important to consider the complexity of data generation techniques since they often require observing classes in multiple domains.

**Multi-domain feature alignment** techniques primarily align features across source-domains through explicit feature alignment. For example, DIRT (Nguyen et al., 2021) aligns transformed domains, CORAL (Sun & Saenko, 2016) and M$^3$SDA (Peng et al., 2019) align second and first-order statistics, MDA (Hu et al., 2019) learn class-wise kernels, and others use measures like Wasserstein distance (Zhou et al., 2021; Wang et al., 2021). Additionally, some methods focus on gradient operations like PGrad (Wang et al., 2023) and Fishr (Rame et al., 2022), while other methods frame it as probabilistic (EQRM in Eastwood et al. (2022)) or causal (CausIRL in Chevalley et al. (2022)) modelling tasks. Other approaches perform domain-discriminative adversarial training (Zhu et al., 2022; Yang et al., 2021; Shao et al., 2019; Li et al., 2018b; Gong et al., 2018). Howerver, since there is a 1:1 correlation between $\mathcal{Y}_L$ classes and their domains, adversarial domain-discriminators may infer an inputs domain from class-discriminative features.

**Meta-learning** approaches improve generalizability by imitating the generalization tasks through meta-train and meta-test objectives; MLDG (Li et al., 2018a) and ARM (Zhang et al., 2021b) are popular base architectures (Zhong et al., 2022; Shu et al., 2021). Other interesting methods perform adversarial training, Transfer (Zhang et al., 2021a), and few-shot learning, FSDG (Qin et al., 2023).

**Contrastive learning** aims is to learn representations, self-supervised (Chen et al., 2020b) or supervised (Khosla et al., 2020; Motiian et al., 2017), such that similar samples are embedded close to each other while distancing dissimilar samples (Huang et al., 2020; Ruan et al., 2022; Kim et al., 2021; Khosla et al., 2020; Chen et al., 2020b; Motiian et al., 2017). These methods primarily focus on multi-domain comparisons in order to identify domain-invariant representations; see Zhou et al. (2023) and the references therein. However, this methodology is insufficient for identifying generalizable representations for domain-linked classes.

**Fairness** notions in DG (Makhlouf et al., 2021) involve reducing the performance discrepancy between protected attributes (e.g. demographic) (Pham et al., 2023; Wang et al., 2020). However, our work enforces fairness to learn generalizable representations for domain-linked classes.

**Research Gap.** Existing DG approaches presuppose all classes are expressed in multiple domains and/or seek to maximize average generalization; thus ignoring the large performance discrepancies between domain-linked and domain-shared classes. Since these domain-linked classes may be of interest in real-world settings, there is a need to understand the factors that impact their performance, and to build models which can improve their generalizability to unseen domains.

## 3 PROBLEM FORMULATION

We formalize some definitions for the purposes of the paper. First, we define a domain as follows.

**Definition 3.1** (Domain). *Let $\mathcal{X}$ denote an nonempty input space (e.g. images, text, etc) and $\mathcal{Y}$ an output label space. We denote as specific domain as $S = \{(\mathbf{x}_j, y_j)\}_{j=1}^n \sim \mathcal{D}^S : \mathcal{X}^S \times \mathcal{Y}^S$, where $\mathbf{x} \in \mathcal{X}^S \subseteq \mathbb{R}^d$ and $y \in \mathcal{Y}^S \subset \mathbb{Z}$.*

Given this definition of a domain, the DG task – which entails learning representations from multiple source-domains to generalize to unseen target-domain(s) – can be formalized as shown below.

**Definition 3.2** (Domain generalization). *Given $K$ training (source) domains $\mathcal{S} = \{S^i \mid i = 1, ..., K\}$ where $S^i = \{(\mathbf{x}_j^i, y_j^i)\}_{j=1}^{n_i}$ denotes the $i$-th source domain with $n_i$ samples, and the joint distributions between each pair of domains are different: $\mathcal{D}^{S^i} \neq \mathcal{D}^{S^j} : 1 \leq i \neq j \leq K$. Then the goal is to learn a predictive function from $\mathcal{S}$ for reliable performance on an unseen, out-of-distribution target-domain $T \sim \mathcal{D}^T : \mathcal{X}^T \times \mathcal{Y}^T$ (i.e. $\mathcal{D}^T \neq \mathcal{D}^{S^i}$ for $i \in \{1, ..., K\}$).*

We evaluated methods for *closed-set* domain generalization (i.e. $\mathcal{Y}^T = \bigcup_{i=1}^K \mathcal{Y}^{S^i}$) where no source-domain expresses all target classes (i.e., $\mathcal{Y}^T \subset \mathcal{Y}^{S^i}$ for $i \in \{1, ..., K\}$). Furthermore, during training there exists a set of classes expressed in only one source-domain, i.e. *domain-linked* classes $\mathcal{Y}_\lambda^S$:

**Definition 3.3** (Set of domain-linked classes). $\mathcal{Y}_\lambda^S = \{y : |\{S \in \mathcal{S}, y \in \mathcal{Y}^S\}| = 1\}$, $\mathcal{Y}_L$ for brevity.

Similarly, classes expressed in multiple domains, i.e. *domain-shared* classes $\mathcal{Y}_\sigma^{S^i}$ as shown below. Here, $\mathcal{Y}^T = \mathcal{Y}_L \cup \mathcal{Y}_S$ and $\mathcal{Y}_L \cap \mathcal{Y}_S = \emptyset$.

**Definition 3.4** (Set of domain-shared classes). $\mathcal{Y}_\sigma^S = \{y : |\{S \in \mathcal{S}, y \in \mathcal{Y}^S\}| > 1\}$, $\mathcal{Y}_S$ for brevity.

**Learning Objective.** The learning objective is to identify a generalizable predictive function $M : \mathcal{X} \rightarrow \mathcal{Y}$ to achieve a minimum predictive error on an unseen, out-of-distribution, test domain $T$ under the previously outlined conditions. The task network is defined as $M = (F \circ G)(\mathbf{x})$, to learn the feature extractor $F : \mathcal{X} \rightarrow \mathcal{H}$ and the classifier $G : \mathcal{H} \rightarrow \mathcal{Y}^\Delta$, where $\Delta$ is a probability simplex.

## 4 Theoretical Analysis

In this section, we theoretically formalize the challenges faced by domain-linked classes by demonstrating the source of performance discrepancies between domain-linked and domain-shared classes. These insights are then used to develop the proposed algorithm in the next section. Our main result comes in the form of the following theorem; details are in App. B.

**Theorem 1.** *Given $K$ source domains $S^i$ for $i \in [1, K]$, each contributing $\alpha_i \cdot m$ i.i.d. samples where $\sum_{i=1}^K \alpha_i = 1$, and each $\alpha_i$ denotes the fraction of samples contributed by a source $S^i$, a target domain $T$, and a hypothesis class $\mathcal{H}$. The error $\epsilon_{T_\ell}(\hat{h})$ of a domain-linked class $\ell$ of hypothesis $\hat{h} \in \mathcal{H}$, an empirical minimizer of $\epsilon(h)$ on the source samples, is upper bounded by*

$$\epsilon_{T_\ell}(\hat{h}) \leq \epsilon_{T_\ell}(h_T^*) + d_{\mathcal{H}\Delta\mathcal{H}}(T, S^\ell) + \mathcal{O}(m^{-1}), \tag{1}$$

*with probability at least $(1 - \delta_\ell)$ for $\delta_\ell \in (0, 1)$, where $\epsilon_{T_\ell}(h_T^*)$ represents the error achieved by $h_T^* = \min_{h \in \mathcal{H}} \epsilon_T(h)$ on the target domain for the class $\ell$. Further, the upper-bound if this were a domain-shared class expressed in $K$ domains is*

$$\epsilon_{T_\ell}(\hat{h}) \leq \epsilon_{T_\ell}(h_T^*) + \sum_{i=1}^K \alpha_i \, d_{\mathcal{H}\Delta\mathcal{H}}(T, S^i) + \mathcal{O}(m^{-1}), \tag{2}$$

*with probability at least $(1 - \delta_s)$ for $\delta_s \in (0, 1)$, where $d_{\mathcal{H}\Delta\mathcal{H}}(\cdot, \cdot)$ is the average distribution divergence between domains. Then, if $d_{\mathcal{H}\Delta\mathcal{H}}(T, S^i) \leq d_{\mathcal{H}\Delta\mathcal{H}}(T, S^\ell)$ for all $i$, the upper-bound in Eq. (1) is greater than that of Eq. (2).*

*Remark 1.* The result shows the dependence on the number of samples. While we use $\mathcal{O}(m^{-1})$ to simplify this dependence, the number of samples for domain-linked classes is usually lower than those for domain shared classes, which is a significant source of the performance discrepancy.

*Remark 2.* The average distribution divergence is another source of performance discrepancy, if the domain-shared classes have a larger and more expressive dataset than domain-linked the condition $d_{\mathcal{H}\Delta\mathcal{H}}(T, S^i) \leq d_{\mathcal{H}\Delta\mathcal{H}}(T, S^\ell)$ is easy to meet. Although not explicitly stated, the diversity of data can be encoded as the span of a given dataset, i.e. the size of the convex hull of all data points.

This theoretical result demonstrates that for a fixed hypothesis class, the distribution divergence and samples play a major role. While the number of samples are usually fixed, we aim to inject the domain diversity from shared classes to improve performance of domain-linked ones. Our empirical result corroborates this theoretical intuition that a high-shared setting, where a higher number of shared classes are observed in different domain indeed helps learning for domain-linked classes.

## 5 Methodology

We introduce the learning algorithm FOND (*Fair and cONtrastive Domain-linked learning*), which learns domain-invariant representations from domain-shared $\mathcal{Y}_S$ classes that improve domain-linked $\mathcal{Y}_L$ class generalization. We achieve this by minimizing the following objective described in Alg. 1:

$$\mathcal{L}_{\text{FOND}} = \mathcal{L}_{task} + \lambda_{xdom} \cdot \mathcal{L}_{xdom} + \lambda_{fair} \cdot \mathcal{L}_{fair}. \tag{3}$$

We impose domain-invariant representation learning by focusing on specific pairwise sample relationships through the contrastive $\mathcal{L}_{xdom}$ objective. Since we require these representations to improve domain-linked class generalizability, we impose *fair representation learning* between $\mathcal{Y}_L$ and $\mathcal{Y}_S$ through the $\mathcal{L}_{fair}$ objective. The following sections describe the formulation these objectives.

---

**Algorithm 1** FOND Training Algorithm

---

**Require:** Source datasets $\mathcal{S}$, feature extractor $F$, projection network $Q$, classification network $G$
1: **while** Not Converged **do**
2:     Sample a batch of data $\mathcal{B} = \{(\mathbf{x}^1, \mathbf{y}^1), (\mathbf{x}^1, \mathbf{y}^1), ..., (\mathbf{x}^K, \mathbf{y}^K)\}$ from all source domains $\mathcal{S}$
3:     $\mathcal{B}_F = \{(\mathbf{h}^1, \mathbf{y}^1), (\mathbf{h}^2, \mathbf{y}^2), ..., (\mathbf{h}^K, \mathbf{y}^K)\} \leftarrow F(\mathcal{B})$     ▷ Generate input representations
4:     $\mathcal{B}_Q = \{(\mathbf{z}^1, \mathbf{y}^1), (\mathbf{z}^2, \mathbf{y}^2), ..., (\mathbf{z}^K, \mathbf{y}^K)\} \leftarrow Q(\mathcal{B}_F)$     ▷ Generate feature projections
5:     $\mathcal{L}_{xdom} \leftarrow \{\mathcal{B}_Q, \alpha, \beta\}$     ▷ Calculate domain-aware loss according to Eq. equation 4
6:     $\mathcal{B}_C \leftarrow G(\mathcal{B}_F)$     ▷ Generate classification logits
7:     $\mathcal{B}_C^{(L)}, \mathcal{B}_C^{(S)} \leftarrow \{\mathcal{B}_C\}$     ▷ Separate logits based on ground-truth label group, i.e. $\mathcal{Y}_L$ or $\mathcal{Y}_S$
8:     $\mathcal{L}_{fair}, \mathcal{L}_{task} \leftarrow \{\mathcal{B}_C^{(L)}, \mathcal{B}_C^{(S)}\}, \{\mathcal{B}_C\}$     ▷ Calculate task and fair losses according to Eq. equation 5
9: **end while**
**return** $F, G$

---

## 5.1 LEARNING DOMAIN-INVARIANT REPRESENTATIONS FROM $\mathcal{Y}_S$ CLASSES

The $\mathcal{Y}_S$ and $\mathcal{Y}_L$ performance discrepancy (Fig. 1b) results since algorithms do not observe enough domain-variances in $\mathcal{Y}_L$ data like they do with $\mathcal{Y}_S$. In alignment with Remark 2, in Sec. 4, observing domain-variances allows models to learn representations that reduce the average distribution divergence between source and target domain data. Therefore, unlike modern feature-alignment DG approaches our learning objective explicitly treats $\mathcal{Y}_L$ and $\mathcal{Y}_S$ class samples differently.

Maximizing the mutual information between positive (same-class) inter-domain samples guides domain-invariant learning (Chen et al., 2020a; Ren et al., 2023). For example, in Fig. 1a an algorithm observing pairwise relationships between samples from domain-shared $\mathcal{Y}_S$ classes may observe that encoding edge features increases mutual information while color reduces it. Furthermore, due the success of "hard negative mining" in representation learning literature (Robinson et al., 2021; Zhang et al., 2022b; Liu et al., 2023), we hypothesize that negative (different-class) intra-domain comparisons are more informative than negative inter-domain comparisons for reducing spurious domain and class correlations. For example, in Fig. 1a, an algorithm may achieve color invariance by minimizing mutual information between samples from different classes (shapes) but from the same domain (color). This is validated in Table. 2 where we observe that variants including both $\alpha$ and $\beta$ ( i.e, FOND and FOND\F) outperform all baselines.

**Motivated approach.** Therefore, we define a feature extractor $F : \mathcal{X} \to \mathcal{H}$ to take a input samples $\mathbf{x} \in \mathcal{X}$ and generate representation vectors, $\mathbf{h} \in \mathcal{H} \subseteq \mathbb{R}^{d_F}$. We regularize the representation vectors by applying a contrastive objective to the output of a projection network $Q : \mathcal{H} \to \mathcal{Z}$ to generate normalized, lower-dimensional, representations $\mathbf{z} \in \mathcal{Z} \subseteq \mathbb{R}^{d_Q}$. The goal of the contrastive objective defined by Eq. (4) is to maximize the cosine similarity of the projected representations $\mathbf{z}$ between same-label samples $y$ (positive pairs) and minimize those that do not (negative pairs).

$$\mathcal{L}_{xdom} = \sum_{i \in I} \frac{-1}{|P(i)|} \sum_{p \in P(i)} \log \frac{\alpha \cdot \exp(\mathbf{z}_i \cdot \mathbf{z}_p / \tau)}{\sum_{j \in I \setminus \{i\}} \beta \cdot \exp(\mathbf{z}_i \cdot \mathbf{z}_j / \tau)}, \tag{4}$$

$$\alpha = \begin{cases} a, & S(\mathbf{z}_i) \neq S(\mathbf{z}_p), \text{ where } a \geq 1 \\ 1, & \text{otherwise} \end{cases}, \beta = \begin{cases} b, & S(\mathbf{z}_i) = S(\mathbf{z}_j), y_i \neq y_j, \text{ where } b \geq 1 \\ 1, & \text{otherwise} \end{cases}$$

Let $i \in I \equiv \{1...N\}$ be the index of a sample (*anchor*) where $N$ denotes the batch size. $P(i) = \{p \in I \setminus \{i\} : y_p = y_i\}$ is the set of indices of all positives in the batch. The term $\alpha$ increases the cosine similarity weight of the anchor $\mathbf{z}_i$ and positive $\mathbf{z}_p$ sample if they are inter-domain ($S(\mathbf{z}_i) \neq S(\mathbf{z}_p)$), pairs. Note that $S(\mathbf{z}_i)$ denotes the domain $S \in \mathcal{S}$ that $\mathbf{z}_i$ belongs to. Additionally, $\beta$ increases the cosine-similarity weight of the anchor $\mathbf{z}_i$ and $\mathbf{z}_j$ if they are negative, intra-domain ( $S(\mathbf{z}_i) = S(\mathbf{z}_p)$), pairs. The FOND\FBA method variant sets $a = b = 1$, FOND\FB sets $a \geq 1, b = 1$ and FOND\F sets $a \geq 1, b \geq 1$; these variants omit fairness (i.e. \F). Note $a$ and $b$ are hyper-parameters.

## 5.2 IDENTIFYING DOMAIN-INVARIANT REPRESENTATIONS FOR $\mathcal{Y}_L$ WITH FAIRNESS

Increasing the weight of positive inter-domain similarity metrics through $\alpha$ in Eq. (4) biases the model towards domain-shared $\mathcal{Y}_S$ generalization since these metrics are not present between domain-linked $\mathcal{Y}_L$ class samples. Consequently, we impose *fairness* to constrain the model to learn generalizabile features from $\mathcal{Y}_S$ classes that are *also* generalizable for $\mathcal{Y}_L$ classes.

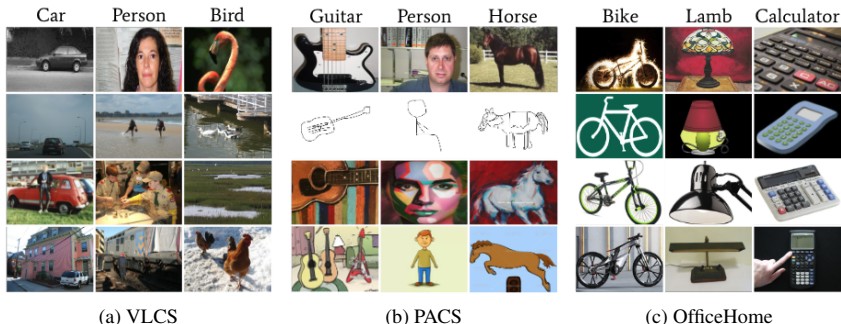

| Car | Person | Bird | Guitar | Person | Horse | Bike | Lamb | Calculator |

| (a) VLCS | (b) PACS | (c) OfficeHome |

Figure 2: **Visualizing domains (rows) and classes (columns) for evaluation dataset.** Note how PACS and OfficeHome exhibit more obvious domain variations than VLCS.

**Applying Fairness for DG.** Notions of fairness in DG require that appropriate statistical measures are equalized across protected attributes (e.g. gender) (Makhlouf et al., 2021). We can formulate these notions of fairness as (conditional) independence statements between random variables: prediction outcome $M(X)$, protected attribute $A$ and class $Y$ (Kilbertus et al., 2017). For example, *demographic parity* $(M(X) \perp A)$ requires the prediction outcomes to be the same across different groups; *equalized odds* $(M(X) \perp A|Y)$ requires that true and false positive rates are the same across different groups; *equalized opportunity* $(M(X) \perp A|Y = y)$ requires that only true positive rates are the same across different groups. However, defining our protected attribute $A$ as whether a sample belongs to $\mathcal{Y}_S$ or $\mathcal{Y}_L$ would make the prediction outcome $M(X)$ completely dependent on $A$.

**Motivated approach.** Therefore, we impose a fairness constraint on the prediction *error rate* such that the domain-invariant representations learned from $\mathcal{Y}_S$ improve the generalizability of the disadvantaged $\mathcal{Y}_L$ classes. The violation of this objective is measured in Eq. (5) by the absolute difference between their classification losses. We observe in Sec. 6.2 that this objective leads to large improvements in $\mathcal{Y}_L$ classes given the model observes a sufficient number of $\mathcal{Y}_S$ classes (*High*-shared).

$$\mathcal{L}_{fair} = |\mathcal{L}_{task}^L - \mathcal{L}_{task}^S|, \text{ where, } \mathcal{L}_{task} = \mathbb{E}_{(\mathbf{x},y) \sim \mathcal{S}} \left[ -\mathbf{y} \cdot \log\left(M(\mathbf{x})\right) \right] \tag{5}$$

## 6 EXPERIMENTS

We specifically focus on analyzing domain-linked class performance under varying 1) shared-class distribution settings (Sec. 6.1), 2) inter-domain variation types, and 3) number of classes. For rigorous, fair and reproducible evaluation, we mirrored the DomainBed (Gulrajani & Lopez-Paz, 2021) test-bed to be consistent with the experimentation in DG literature.

We report class-averaged classification accuracy and standard errors for $\mathcal{Y}_L$ classes here. Reported values arising from three Monte-Carlo runs for three datasets, across four domains, for five random hyper-parameter selections, and for the *Low* and *High*-shared settings. Overall, our extensive experiments summarize the results of 360 experiments per baseline. We focus on the high-shared setting for $\mathcal{Y}_L$ in the main paper, results for $\mathcal{Y}_S$, low-shared setting are in the App. C. Implementation details and the code is provided in App. A and supplementary, respectively.

### 6.1 EXPERIMENTAL SETUP

**Datasets.** To evaluated $\mathcal{Y}_L$ performance with respect to inter-domain variations and the number of classes, we required keeping consistent a) dataset sizes and b) number of domains. Therefore we chose DG literature gold standard datasets (Zhou et al., 2023): PACS, VLCS, and OfficeHome, as

Table 1: **Evaluation dataset properties.** This table characterizes the datasets and defines the number of $\mathcal{Y}_L$ versus $\mathcal{Y}_S$ classes for the *Low* and *High* class distribution experimental settings.

| | | | Properties | | Settings ($|\mathcal{Y}_S| : |\mathcal{Y}_L|$) | |
| --- | --- | --- | --- | --- | --- | --- |
| **Datasets** | **# Domains** | **# Classes** | **# Samples** | **Domain Variation Type** | **Low-shared** | **High-shared** |
| PACS | 4 | 7 | 9,991 | Style-Based | 3:4 | 5:2 |
| VLCS | 4 | 5 | 10,729 | Real-World | 2:3 | 4:1 |
| OfficeHome | 4 | 65 | 15,588 | Style-Based | 25:40 | 50:15 |

Table 2: **High setting out-of-distribution test accuracy (%) for $\mathcal{Y}_L$ classes.**

| Algorithm | VLCS | PACS | OfficeHome | Average | % Improvement |
|---|---|---|---|---|---|
| ERM | $51.8 \pm 3.3$ | $14.7 \pm 2.2$ | $37.5 \pm 0.6$ | 34.6 | *(0.0%)* |
| CORAL (Sun & Saenko, 2016) | $49.8 \pm 4.2$ | $13.7 \pm 1.0$ | $38.9 \pm 0.2$ | 34.1 | *(-1.4%)* |
| MLDG (Li et al., 2018a) | $45.2 \pm 3.4$ | $13.8 \pm 0.5$ | $37.4 \pm 0.7$ | 32.1 | *(-7.2%)* |
| ARM (Zhang et al., 2021b) | $49.0 \pm 1.4$ | $16.2 \pm 2.9$ | $38.4 \pm 0.2$ | 34.5 | *(-0.3%)* |
| SelfReg (Kim et al., 2021) | $41.9 \pm 0.2$ | $13.4 \pm 1.2$ | $39.5 \pm 0.6$ | 31.6 | *(-8.7%)* |
| CAD (Ruan et al., 2022) | $51.7 \pm 5.8$ | $13.1 \pm 0.7$ | $36.4 \pm 1.4$ | 33.7 | *(-2.6%)* |
| Transfer (Zhang et al., 2021a) | $48.9 \pm 3.0$ | $16.0 \pm 1.6$ | $36.8 \pm 0.2$ | 33.9 | *(-2.0%)* |
| CausIRL (Chevalley et al., 2022) | $48.9 \pm 2.5$ | $13.3 \pm 1.5$ | $39.2 \pm 0.2$ | 33.8 | *(-2.3%)* |
| EQRM (Eastwood et al., 2022) | $45.4 \pm 3.5$ | $\underline{17.9 \pm 2.0}$ | $37.8 \pm 0.1$ | 33.7 | *(-2.6%)* |
| PGrad (Wang et al., 2023) | $40.2 \pm 1.8$ | $\underline{12.6 \pm 1.4}$ | $39.0 \pm 0.8$ | 30.6 | *(-11.6%)* |
| FOND | $\mathbf{72.1 \pm 3.5}$ | $\mathbf{19.1 \pm 0.6}$ | $40.6 \pm 0.4$ | $\mathbf{43.9}$ | *(+26.9%)* |
| FOND\F | $51.7 \pm 6.0$ | $17.5 \pm 1.4$ | $\underline{40.8 \pm 0.6}$ | $\underline{36.7}$ | *(+6.1%)* |
| FOND\FB | $44.0 \pm 2.3$ | $15.4 \pm 0.6$ | $\mathbf{41.7 \pm 0.7}$ | 33.7 | *(-2.6%)* |
| FOND\FBA | $51.3 \pm 2.8$ | $17.3 \pm 1.3$ | $39.1 \pm 0.5$ | 35.9 | *(+3.8%)* |

shown in Table 1 and Fig. 2. While PACS (Fig. 2b) and OfficeHome (Fig. 2c) datasets share similar style-based domain-variations, there is a $\sim$10x difference in class size. Furthermore, although VLCS (Fig. 2a) and PACS have similar class sizes, VLCS expresses nuanced real-world domain-variations.

**Defining Shared-Class Distribution Settings.** We define two shared-class distribution settings – *Low* and *High* – denoting the relative number of shared classes $|\mathcal{Y}_S|$ with respect to the total $|\mathcal{Y}_T|$ (refer to Table 1). In the *Low* setting $\sim 1/3$ of the classes are domain-shared; $\sim 2/3$ in the *High* setting. $\mathcal{Y}_L$ and $\mathcal{Y}_S$ classes were randomly selected and assigned round-robin to each source-domain.

**Baselines.** Baselines were selected to a) cover a variety of foundational DG methodologies, that are b) well represented in DG literature and have c) been benchmarked against DomainBed (Gulrajani & Lopez-Paz, 2021). Therefore, we evaluate: naive empirical risk minimization, ERM; popular distribution-alignment, CORAL (Sun & Saenko, 2016); contrastive mixing-based, RSC (Huang et al., 2020) predecessor, SelfReg (Kim et al., 2021); contrastive CLIP-based (Radford et al., 2021) method, CAD (Ruan et al., 2022); popular meta-learning baselines, ARM (Zhang et al., 2021b) and MLDG (Li et al., 2018a); adversarial meta-learning network, Transfer (Zhang et al., 2021a); causal representations, CausIRL (CORAL variant) (Chevalley et al., 2022); gradient optimization, PGrad (Wang et al., 2023); probabilistic framework, EQRM (Eastwood et al., 2022).

## 6.2 EXPERIMENTAL RESULTS

In this section, we present the experimental results and empirically demonstrate that FOND benefits the domain-linked class performance ($\mathcal{Y}_L$) by transferring domain-invariant representations learned from domain-shared classes ($\mathcal{Y}_S$).

**Effect of shared-class settings and total class size.** We report the performance under the low-and high-shared setting in Fig. 3 (Table 4 in App. C shows detailed listings). FOND relies on observing domain-shared classes, therefore it only demonstrates top-4 performance in the *Low* setting. In the *High* setting we observe that FOND consistently outperforms all baselines as shown in Table 2. Strikingly, FOND results in a 39% performance improvement over the best baseline (ERM) for VLCS. There are some interesting trends in Fig. 3. It seems that while for VLCS and OfficeHome the performance improves from *Low* to *High*, this is not true for PACS. Does this mean that shared setting does not help? To probe this further, in Fig. 4 we track only those classes which were domain-linked

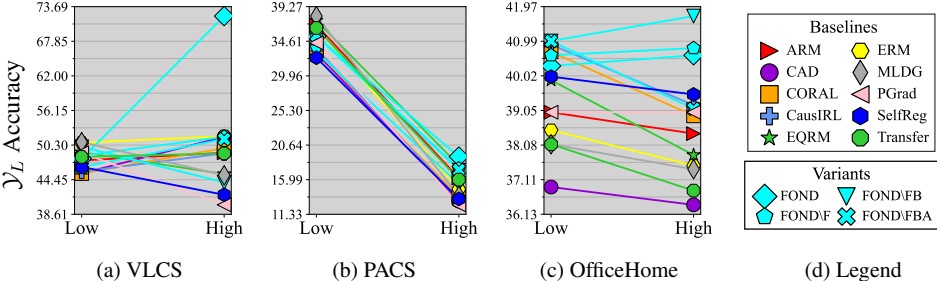

Figure 3: **Average $\mathcal{Y}_L$ classes performances for *Low* and *High* shared-class distribution settings.**

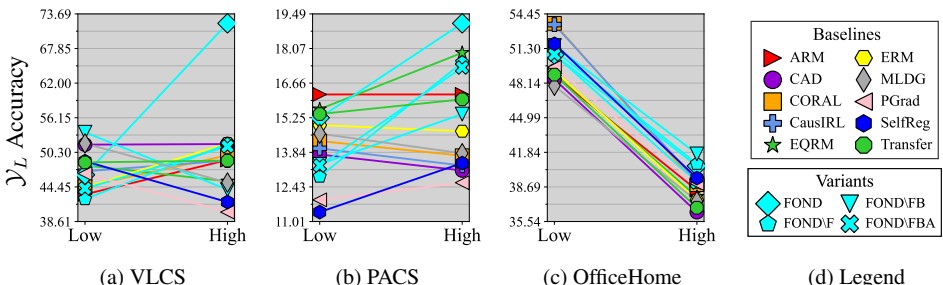

**Figure 4:** **Tracking individual domain-linked $\mathcal{Y}_L$ classes performances for classes in both *Low* and *High* shared-class distribution settings.** We track the changes in the domain-linked classes present in both the *Low* and *High* setting.

in both *Low* and *High* setting to see how an increase in domain-shared classes impacts the performance of domain-linked classes. We see that in fact for both VLCS and PACS FOND improves the performance for domain-linked classes! But another interesting trend presents itself for Office-Home. We observe that while FOND outperforms all other baselines in the *High* setting, OfficeHome domain-linked classes performance in Fig.4(c) decreased. To get full picture, in Fig.3(c) we note that the performance gains are modest, this indicates that *Low* and *High* are not relative but absolute. In other words, a higher number of domain-linked classes require a large number of domain-shared classes, which again underscores the importance of domain-shared classes.

**Effect of inter-domain variations.** To analyze the impact of domain differences we turn to characteristics of the dataset. As shown in Fig. 2c, VLCS domains are real-world while for PACS and OfficeHome domain differences are more obvious. We find while FOND outperforms all baselines for each of the datasets (Table 2), the results for VLCS are the most prominent. In fact, each baseline struggles on PACS and OfficeHome. Our theoretical analysis sheds light on this trend. Specifically, from Thm. 1 we see that the distribution divergence $d_{\mathcal{H}\Delta\mathcal{H}}(T, S^\ell)$ between the source and the target plays a major role for effective transfer, and puts a fundamental limit on the performance. Interestingly, FOND is able to leverage domain-shared classes to maximize performance.

**Impact on domain-shared classes.** While the goal of this work is to improve domain-linked performance, a natural question arises regarding the impact of this prioritization on the domain shared classes. To analyze this performance, in Fig. 7 we present a comparative analysis of the $\mathcal{Y}_L$ vs. $\mathcal{Y}_S$ accuracies. For each of the dataset, FOND results in striking improvement in $\mathcal{Y}_L$ accuracies as compared to all the baselines. We also observe how different FOND variants can be used to reconcile the trade-off in the real-world. Interestingly, the performance improvements for OfficeHome show how modest loss of $\mathcal{Y}_S$ can lead to significant gains in $\mathcal{Y}_L$, highlighting the strenghts of FOND.

**Visualizing learned representations.** To further understand the representations learned by FOND, for each dataset we visualize its learned latent representations and compare them to those learned by ERM (top-performing-baseline in Table 2). For ERM, the class-colored clusters are also distinctly sub-clustered by domain (e.g. broken circle in Fig. 6a and Fig. 6b). For FOND, class-colored clusters are not as distinctly sub-clustered by domain (Fig. 6d and Fig. 6e); therefore FOND learns more domain-invariant representations. Furthermore, for domain-linked $\mathcal{Y}_L$ class samples, FOND (e.g. solid circle in Fig. 6f) yields more generalizable representations than ERM (Fig. 6c. Finally, while all methods struggle on the pink class, FOND empirically maintains top performance.

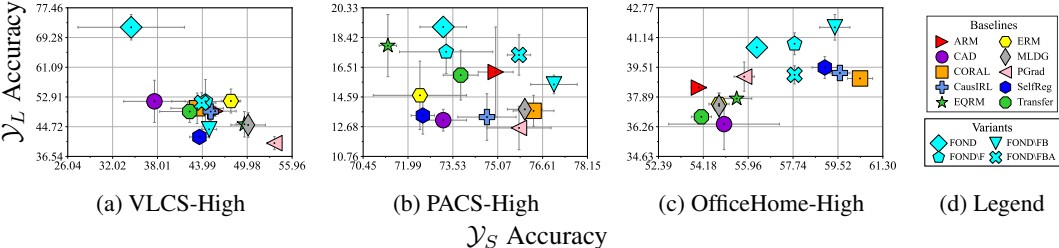

**Figure 5:** **Visualizing $\mathcal{Y}_L$ and $\mathcal{Y}_S$ performance trade-offs on the *High* shared-class distribution setting.** Our method outperforms all baselines on $\mathcal{Y}_L$ classes with more competitive $\mathcal{Y}_S$ class performance as the total number of target classes increases (left to right). Additional plots in App. C.

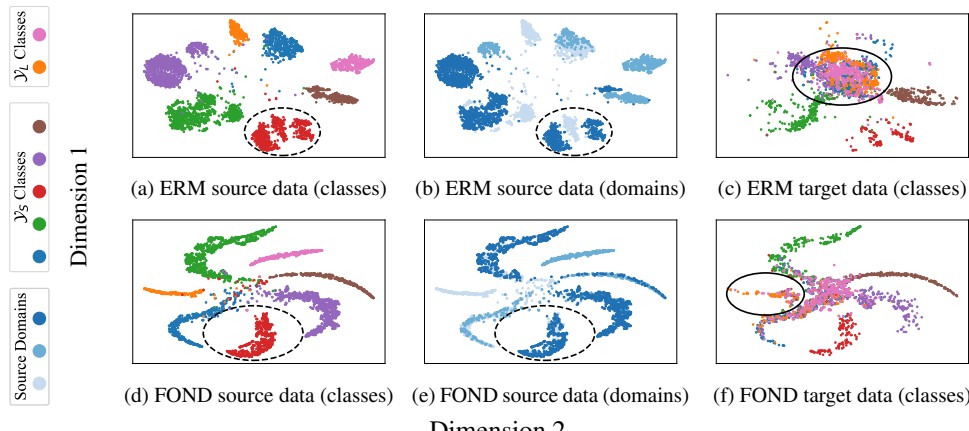

Figure 6: **t-SNE latent representation visualization for the PACS-*High* dataset**. Each row visualizes the representations of FOND versus top-performing baseline ERM. Source-domain (*Photo*, *Art* and *Sketch*) representations are colored by class (left) and domain (center). Target-domain (*Cartoon*) representations are colored by class (right). Refer to the analysis of the domain-linked $\mathcal{Y}_L$ class generalization (solid circle) and domain-invariant learning (broken circle) in Sec. 6.2.

## 7 DISCUSSION

**Summary.** Domain generalization (DG) in real-world settings often suffers from data scarcity leading to classes which are only observed in specific domains, i.e. they are *domain-linked*. DG efforts primarily focus on improving the overall accuracy which can lead to critical failures in the real-world. Motivated from this challenge, we focused on improving the out-of-distribution generalization for domain-linked classes by transferring domain-invariant knowledge from *domain-shared* classes. Consequently, we proposed FOND; the first method for domain-linked domain generalization, which achieves state-of-the-art performance as compared to the current SOTA DG approaches.

**Significance.** By introducing the domain-linked DG task we draw attention to the large performance discrepancies between domain-linked and domain-shared classes, across current state-of-the-art DG algorithms. A similar challenge has been observed in the context of data-scarcity in long-tailed classification ($50 - 1000$ classes) Gu et al. (2022), but we show that DG models struggle even for a modest number of classes, highlighting a need to develop methods that can address this challenging scenario. Our focus here is only on domain-linked class performance. This is not a limitation, but a change in focus. We also observe a need to develop insightful metrics to understand the DG representations, since a sole focus on overall accuracy can be misleading. Finally, we also note that DG methods implicitly assume that training domains are *different*, FOND is the only method that is able to achieve reliable performance under nuanced real-world domain variations (e.g. VLCS).

**Key Takeaways.** From our analysis, we observe that the proposed method works well in the *High*-shared setting. This highlights the need for observing a sufficient number of domain-shared classes in multiple domains to be able to draw useful representations. We also note that the performance differences between *Low* and *High* are especially stark for VLCS and PACS, which have lower number of classes as compared to OfficeHome. This indicates that *Low* and *High* are not dependent on proportions of domain-linked vs. domain-shared classes, but their absolute number. This also presents an opportunity for rethinking DG for small vs. large class settings. More importantly, guided by our theoretical insights which demonstrates the role of average distributional divergence between source and target domains, we observe that FOND performs the best on VLCS, which contains real-world domain shifts. The ability to leverage knowledge from domain-shared classes to accomplish state-of-the-art results for domain-linked ones opens tremendous possibilities for real-world DG, stimulating domain generalization research for real-world data-scarce domains.

**Limitations and future work.** Like all baselines, FOND also struggles in the *Low*-shared setting, which is due to the fact the there aren't enough shared classes to learn from. Next, we make a binary distinction between domain-linked and domain-shared classes, but it will be exciting to explore methods that can adapt to varying levels of shared-*ness* in future. Nevertheless, our work calls for paying a closer attention to the biases and assumptions in DG.

## 8 ETHICS STATEMENT

Our fairness objective is conditioned on a class being represented in multiple domains, which in the real-world may result from representation inequalities of protected attributes and/or classes which are only observed in certain domains (or rarely observed in others). Therefore, careful consideration is required when deploying fairness based DG research since they could make decisions that unfairly impact specific groups. This work demonstrates how we can begin to think about these challenging tasks. On the computation front, domain generalization research is computationally heavy since it requires multiple validation cycles for each dataset, algorithm, hyper-parameter search space and shared-class distribution setting. Therefore, as we expand DG research we need to improve ML resource efficiency to both increases its accessibility and reduce negative environmental consequences.

## 9 REPRODUCIBILITY

For reproducability, all algorithms, datasets, hyper-parameter searching and model selection follow the DomainBed (Gulrajani & Lopez-Paz, 2021) benchmark standard; refer to App. A. All code is self contained and the corresponding training scripts are provided in the supplementary resources.

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

# A EXPERIMENTAL SETUP

## A.1 IMPLEMENTATION DETAILS

For consistency, all algorithms have a fine-tuned ResNet-18 backbone (He et al., 2016) pre-trained on ImageNet (Deng et al., 2009). Specifically, we replace the final (softmax) layer, insert a dropout layer and then fine-tune the entire network. Since minibatches from different domains follow different distributions, batch normalization degrades domain generalization algorithms (Seo et al., 2019). Therefore, we freeze all batch normalization layers before fine-tuning. Additionally, the training data augmentations are: random size crops and aspect ratios, resizing to 224 × 224 pixels, random horizontal flips, random color jitter, and random gray-scaling. Our experiments ran on different GPUs: NVIDIA RTXA6000, NVIDIA GeForce RTX2080. Further implementation details are provided in the attached code.

## A.2 HYPER-PARAMETER SEARCH

For each algorithm we perform five random search attempts over a joint distributions of all their hyper-parameters. The performance of each hyper-parameter is evaluated using the strategy outlined in App. A.3. This is repeated for each of the five sets of hyper-parameters and the set maximizing the average domain-linked $\mathcal{Y}_L$ accuracy is selected. This search is performed for across three different seeds where all hyper-parameters are optimized anew for each algorithm, dataset and partial-overlap setting. The hyper-parameter search space for each algorithm is provided in the attached code.

## A.3 MODEL SELECTION

Given $K$ domains, we train $K$ models, sharing the same hyper-parameters $\theta$, but each model holds a different domain out. During the training of each model, 80% of the training data from each domain is used for training and the other 20% is used to determine the version that will be evaluated. We evaluate each model on its held-out domain data, and average the $\mathcal{Y}_L$ accuracy of these $K$ models over their held-out domains. This provides us with an estimate of the quality of a given set of hyper-parameters. This strategy was chosen because it aligns with the goal of maximizing expected performance under out-of-distribution domain-variation without picking the model using the out-of-distribution data. The $\mathcal{Y}_L$ accuracy performance across held-out domains and final averages for each dataset, algorithm and partial-overlap setting are displayed in Table 4.

## A.4 DATASETS

**PACS** (Li et al., 2017) is a 9,991-image dataset consisting of four domains corresponding to four different image styles: photo (P), art-painting (A), cartoon (C) and sketch (S). Each of the four domains hold seven object categories: dog, elephant, giraffe, guitar, horse, house and person.

**VLCS** (Fang et al., 2013) is a 10,729-image dataset consisting of four domains corresponding to four different datasets: VOC2007 (V), LabelMe (L), Caltech101 (C) and SUN09 (S). Each of the four domains hold five object categories: bird, car, chair, dog and person.

**OfficeHome** (Venkateswara et al., 2017) is a 15,588-image dataset consisting of images of everyday objects organized into four domains; art-painting, clip-art, images without backgrounds and real-world photos. Each of the domains holds 65 object categories typically found in offices and homes.

Table 3: **Domain-shared ($\mathcal{Y}_S$) classes for each dataset and shared-class distribution setting.** The left table defines the number of $\mathcal{Y}_S$ classes and the right displays the corresponding $\mathcal{Y}_L$ classes.

| | $|\mathcal{Y}_S|/|\mathcal{Y}_T|$ | | | $\mathcal{Y}_L$ | | |
|---|---|---|---|---|---|---|
| Setting | PACS | OfficeHome | VLCS | PACS | OfficeHome | VLCS |
| Low | 3/7 | 25/65 | 2/5 | {0,1,3,5,6} | {0-13, 30-34, 44-64} | {0,1,4} |
| High | 5/7 | 50/65 | 4/5 | {1,6} | {0-4, 44-64} | {1} |

A.5    MODEL ARCHITECTURE

In this section we describe the FOND architecture components and outline the intermediate latent representations that are used for our learning objectives.

- The *Feature Extraction Network*, $F(.)$, takes a training input sample $\mathbf{x} \in \mathcal{S}$ and generates a representation vector, $\mathbf{h} = F(\mathbf{x}) \in \mathbb{R}^{d_F}$ where $d_F = 512$.
- The *Projection Network*, $Q(.)$, takes the representation vector $\mathbf{h}$ and non-linearly projects it to a lower-dimensional vector $\mathbf{z} = P(\mathbf{h}) \in \mathbb{R}^{d_Q}$ where $d_Q = 256$. Additionally, the projection vector is normalized $||\mathbf{z}|| = 1$. These projections are used for FONDs''s contrastive learning objective Eq. (4).
- The *Classification Network*, $G(.)$, performs the image classification downstream task with the representations generated by $F(.)$, i.e., $\mathbf{h} \in \mathbb{R}^{d_F}$. The network's output is a vector of dimension $|\mathcal{Y}_T|$ denoting the softmax label probabilities of the input $\mathbf{x}$.

# B    PROOF OF THEORETICAL RESULT

**Proof.** Given $K$ source domains and a class $c \in [C]$, we use the result from Thm. 4 in Ben-David et al. (2010) assuming that each of the domains contributes $\alpha_i \cdot m$ i.i.d. samples where $\sum_{i=1}^{K} \alpha_i = 1$ without loss of generality. The following result is reproduced from Ben-David et al. (2010), where we have used the Landau notation to simplify the result. Additionally, for our analysis, $\lambda$-closeness ($\lambda_\alpha = 0$) is assumed because of the extremely large capacities of neural-networks; this assumption is often made in DG and is experimentally validated (Vedantam et al., 2021). Furthermore, since $\epsilon_T(h_T^*)$ is fixed, our analysis focuses on the $d_\alpha$ and $O(m^{-1})$ terms.

**Theorem 2** (Ben-David et al. (2010) Thm. 4). *Given $K$ source domains $S^i$ for $i \in [1, K]$, each contributing $\alpha_i \cdot m$ i.i.d. samples where $\sum_{i=1}^{K} \alpha_i = 1$, and each $\alpha_i$ denotes the fraction of samples contributed by a source $S^i$, a target domain $T$, that the hypothesis class $\mathcal{H}$ needs to generalize to. Then, if $\epsilon_{T_c}(h_T^*)$ represents the error achieved by hypothesis $h_T^* = \min_{h \in \mathcal{H}} \epsilon_T(h)$ on a class $c \in [1, C]$ on the target domain, the error $\epsilon_{T_c}(\hat{h})$ of $c$ on the target domain $T$ using hypothesis $\hat{h} \in \mathcal{H}$, an empirical minimizer of $\epsilon(h)$ on the source samples is upper bounded by*

$$\epsilon_{T_c}(\hat{h}) \leq \epsilon_{T_c}(h_T^*) + \sum_{i=1}^{K} \alpha_i \, d_{\mathcal{H}\Delta\mathcal{H}}(T, S^i) + \mathcal{O}(m^{-1}) \tag{6}$$

*with probability at least $(1 - \delta)$, where $d_{\mathcal{H}\Delta\mathcal{H}}(\cdot, \cdot)$ is the average distribution divergence between domains, and $\delta \in (0, 1)$.*

Now, let $\ell$ and $s$ denote a domain-linked and domain-shared class, respectively. We will be using Thm. 2 to evaluate the upper-bound on error for $\ell$ and $s$.

For class $\ell$, there is just once source domain therefore $\alpha_i = 1$, and we have

$$\epsilon_{T_\ell}(\hat{h}) \leq \epsilon_{T_\ell}(h_T^*) + d_{\mathcal{H}\Delta\mathcal{H}}(T, S^\ell) + \mathcal{O}(m^{-1}),$$

with probability at least $(1 - \delta_\ell)$ for $\delta_\ell \in (0, 1)$, where $\epsilon_{T_\ell}(h_T^*)$ represents the error achieved by $h_T^* = \min_{h \in \mathcal{H}} \epsilon_T(h)$ on the target domain for the class $\ell$.

Now, of the class $\ell$ is expressed in $K$ domains, then the upper-bound from Thm. 2 is given by

$$\epsilon_{T_{ell}}(\hat{h}) \leq \epsilon_{T_\ell}(h_T^*) + \sum_{i=1}^{K} \alpha_i \, d_{\mathcal{H}\Delta\mathcal{H}}(T, S^i) + \mathcal{O}(m^{-1}),$$

Analyzing the upper-bounds above we note that for the case when $\ell$ is domain-shared class, if any of the source domains $S^i$ are closer to the target domain $T$ as compared to when it is domain-linked class i.e.

$$d_{\mathcal{H}\Delta\mathcal{H}}(T, S^i) \leq d_{\mathcal{H}\Delta\mathcal{H}}(T, S^\ell) \, \forall \, i,$$

the upper-bound for $\epsilon_{T_\ell}(\hat{h})$ is greater than $\epsilon_{T_s}(\hat{h})$. The bounds are equal when $d_{\mathcal{H}\Delta\mathcal{H}}(T, S^i) = d_{\mathcal{H}\Delta\mathcal{H}}(T, S^\ell)$ for all $i$.

# C  ADDITIONAL RESULTS

Table 4: **Results on $\mathcal{Y}_L$ class accuracy evaluated on PACS, VLCS and OfficeHome for the** *Low* **and** *High* **shared-class distribution settings.** FOND and variants significantly outperform all baselines during the *High* setting with top-4 performance on *Low* and best performance overall.

| Setting | Algorithm | Datasets | | | |
|---|---|---|---|---|---|
| | | VLCS | PACS | OfficeHome | Average |
| | ERM | 50.7 ± 1.0 | 36.5 ± 0.5 | 38.5 ± 0.4 | 41.9 *(0.0)* |
| | CORAL (Sun & Saenko, 2016) | 45.5 ± 1.6 | 33.3 ± 0.8 | 40.7 ± 0.2 | 39.8 *(-2.1)* |
| | MLDG (Li et al., 2018a) | **50.8 ± 2.0** | **38.0 ± 0.1** | 38.1 ± 0.1 | 42.3 *(+0.4)* |
| | ARM (Zhang et al., 2021b) | 47.7 ± 0.9 | 36.8 ± 1.3 | 39.0 ± 0.1 | 41.2 *(-0.7)* |
| | SelfReg Kim et al. (2021) | 46.6 ± 1.2 | 32.4 ± 0.4 | 40.0 ± 0.3 | 39.6 *(-2.3)* |
| Low | CAD (Ruan et al., 2022) | 45.5 ± 1.5 | 33.0 ± 0.9 | 36.9 ± 1.2 | 38.5 *(-3.4)* |
| | Transfer (Zhang et al., 2021a) | 48.3 ± 0.6 | 36.4 ± 1.8 | 38.1 ± 0.3 | 40.9 *(-1.0)* |
| | CausIRL (Chevalley et al., 2022) | 45.8 ± 0.9 | 33.6 ± 0.9 | **40.9 ± 0.2** | 40.1 *(-1.8)* |
| | EQRM (Eastwood et al., 2022) | 49.1 ± 1.1 | 37.4 ± 0.7 | 39.9 ± 0.2 | 42.1 *(+0.2)* |
| | PGrad (Wang et al., 2023) | 49.0 ± 0.7 | 34.4 ± 0.5 | 39.0 ± 0.3 | 40.8 *(-1.1)* |
| | FOND | 48.0 ± 0.4 | 35.3 ± 1.2 | 40.3 ± 0.3 | 41.2 *(-0.7)* |
| | FOND\F | 48.5 ± 1.0 | 35.3 ± 0.5 | 40.6 ± 0.6 | 41.5 *(-0.4)* |
| | FOND\FB | 50.0 ± 0.2 | 33.2 ± 0.5 | 41.0 ± 0.5 | 41.4 *(-0.5)* |
| | FOND\FBA | 46.6 ± 1.0 | 35.4 ± 1.2 | **41.0 ± 0.4** | 41.0 *(-0.9)* |
| | ERM | 51.8 ± 3.3 | 14.7 ± 2.2 | 37.5 ± 0.6 | 34.6 *(0.0)* |
| | CORAL (Sun & Saenko, 2016) | 49.8 ± 4.2 | 13.7 ± 1.0 | 38.9 ± 0.2 | 34.1 *(-0.5)* |
| | MLDG (Li et al., 2018a) | 45.2 ± 3.4 | 13.8 ± 0.5 | 37.4 ± 0.7 | 32.1 *(-2.5)* |
| | ARM (Zhang et al., 2021b) | 49.0 ± 1.4 | 16.2 ± 2.9 | 38.4 ± 0.2 | 34.5 *(-0.1)* |
| | SelfReg (Kim et al., 2021) | 41.9 ± 0.2 | 13.4 ± 1.2 | 39.5 ± 0.6 | 31.6 *(-3.0)* |
| | CAD (Ruan et al., 2022) | 51.7 ± 5.8 | 13.1 ± 0.7 | 36.4 ± 1.4 | 33.7 *(-0.9)* |
| High | Transfer (Zhang et al., 2021a) | 48.9 ± 3.0 | 16.0 ± 1.6 | 36.8 ± 0.2 | 33.9 *(-0.7)* |
| | CausIRL (Chevalley et al., 2022) | 48.9 ± 2.5 | 13.3 ± 1.5 | 39.2 ± 0.2 | 33.8 *(-0.8)* |
| | EQRM (Eastwood et al., 2022) | 45.4 ± 3.5 | 17.9 ± 2.0 | 37.8 ± 0.1 | 33.7 *(-0.9)* |
| | PGrad (Wang et al., 2023) | 40.2 ± 1.8 | 12.6 ± 1.4 | 39.0 ± 0.8 | 30.6 *(-4.0)* |
| | FOND | **72.1 ± 3.5** | **19.1 ± 0.6** | 40.6 ± 0.4 | **43.9 *(+9.3)*** |
| | FOND\F | 51.7 ± 6.0 | 17.5 ± 1.4 | 40.8 ± 0.6 | 36.7 *(+2.1)* |
| | FOND\FB | 44.0 ± 2.3 | 15.4 ± 0.6 | **41.7 ± 0.7** | 33.7 *(-0.9)* |
| | FOND\FBA | 51.3 ± 2.8 | 17.3 ± 1.3 | 39.1 ± 0.5 | 35.9 *(+1.3)* |
| | ERM | 51.3 ± 2.2 | 25.6 ± 1.4 | 38.0 ± 0.2 | 38.3 *(0.0)* |
| | CORAL (Sun & Saenko, 2016) | 47.7 ± 2.9 | 23.5 ± 0.9 | 39.8 ± 0.2 | 37.0 *(-1.3)* |
| | MLDG (Li et al., 2018a) | 48.0 ± 2.7 | 25.9 ± 0.3 | 37.8 ± 0.4 | 37.2 *(-1.1)* |
| | ARM (Zhang et al., 2021b) | 48.4 ± 1.2 | 26.5 ± 2.1 | 38.7 ± 0.2 | 37.9 *(-0.4)* |
| Low/High | SelfReg (Kim et al., 2021) | 44.3 ± 0.7 | 22.9 ± 0.8 | 39.8 ± 0.5 | 35.6 *(-2.7)* |
| | CAD (Ruan et al., 2022) | 48.6 ± 3.7 | 23.1 ± 0.8 | 36.7 ± 1.3 | 36.1 *(-2.2)* |
| | Transfer (Zhang et al., 2021a) | 48.6 ± 1.8 | 26.2 ± 1.7 | 37.5 ± 0.3 | 37.4 *(-0.9)* |
| | CausIRL (Chevalley et al., 2022) | 47.4 ± 1.7 | 23.5 ± 1.2 | 40.1 ± 0.2 | 37.0 *(-1.3)* |
| | EQRM (Eastwood et al., 2022) | 47.3 ± 2.3 | **27.7 ± 1.4** | 38.9 ± 0.2 | 37.9 *(-0.3)* |
| | PGrad (Wang et al., 2023) | 44.6 ± 1.3 | 23.5 ± 1.0 | 39.0 ± 0.6 | 35.7 *(-2.6)* |
| | FOND | **60.1 ± 2.0** | 27.2 ± 0.9 | 40.5 ± 0.4 | **42.6 *(+4.3)*** |
| | FOND\F | 50.1 ± 3.5 | 26.4 ± 1.0 | 40.7 ± 0.6 | 39.1 *(+0.9)* |
| | FOND\FB | 47.0 ± 1.3 | 24.3 ± 0.6 | **41.4 ± 0.6** | 37.6 *(-0.7)* |
| | FOND\FBA | 49.0 ± 1.9 | 26.4 ± 1.3 | 40.1 ± 0.5 | 38.5 *(+0.2)* |

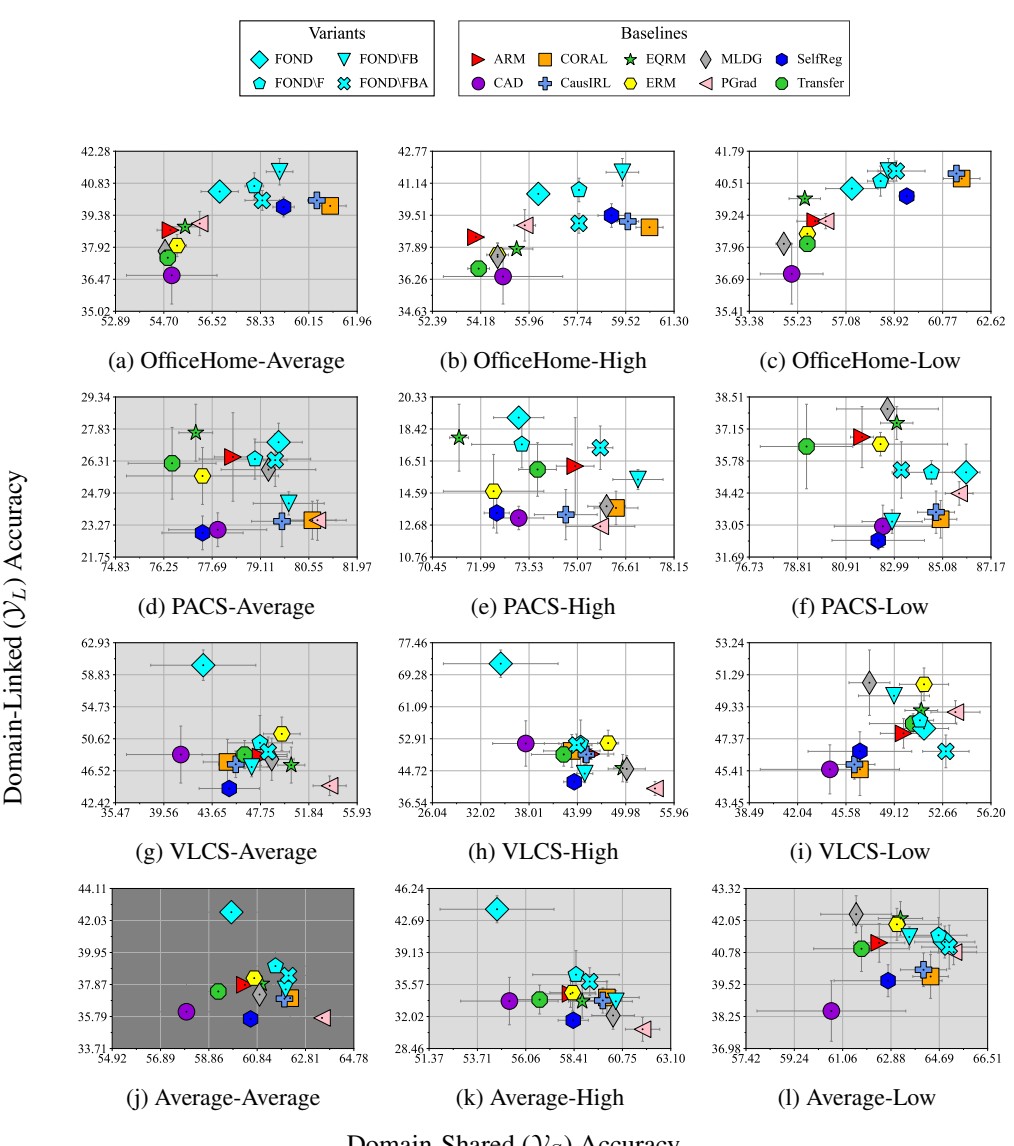

Figure 7: **Visualizing baseline and ablation algorithm accuracy between $\mathcal{Y}_L$ and $\mathcal{Y}_S$ classes across all datasets and shared-class settings.** The white plots communicate accuracies for each dataset's *High* and *Low* shared-class settings. The light-grey plots communicate average accuracies for each dataset (left-most column) and shared-class setting (bottom-row). The dark-grey plot (bottom-left) communicates the average accuracies across all datasets and shared-class settings. The exact values and standard error bars are displayed in Table 4.

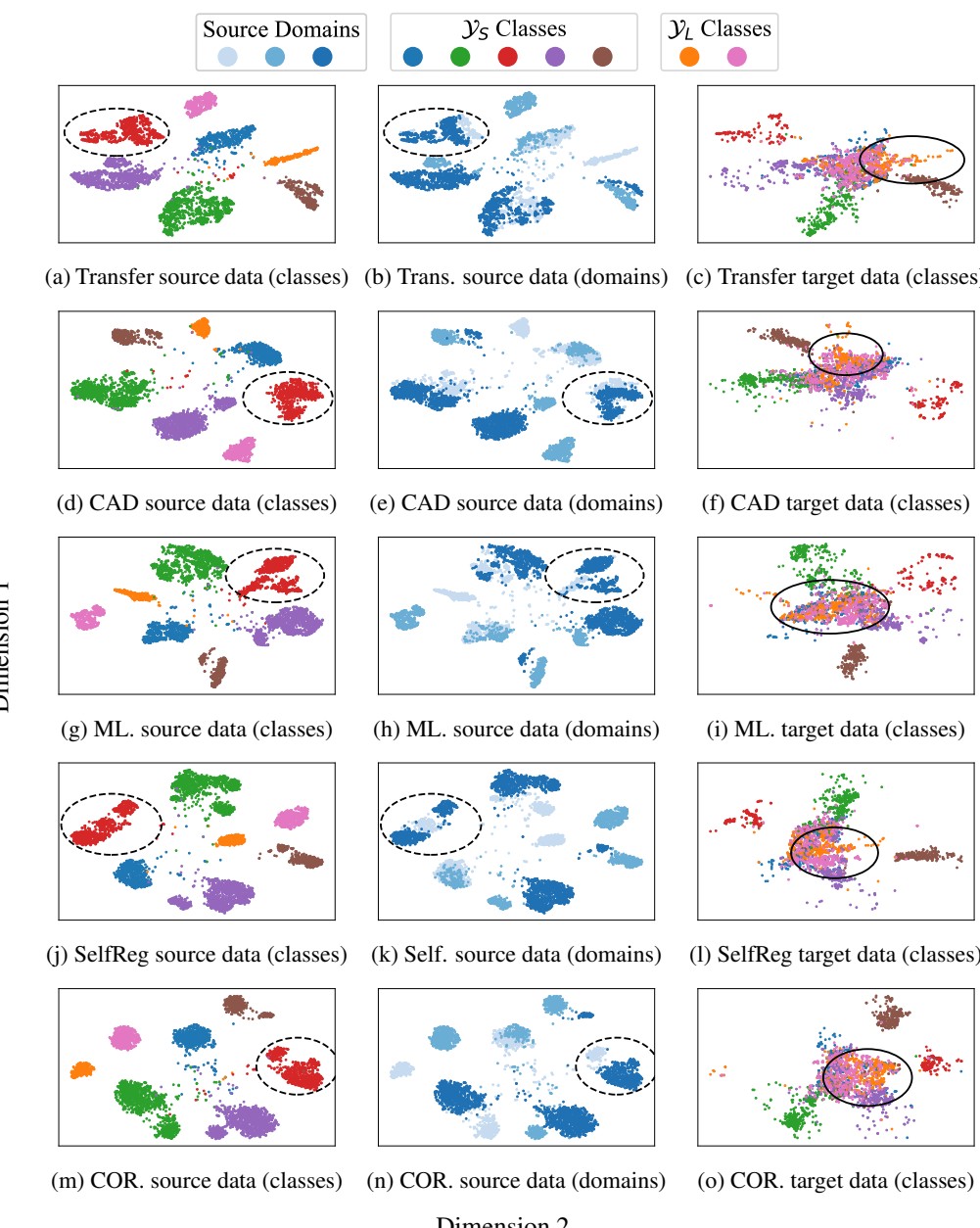

Figure 8: **Additional t-SNE latent representation visualization for the PACS-*High* dataset**. Each row visualizes the additional representations of the baseline algorithms (i.e. Transfer, CAD, MLDG, SelfReg, CORAL). Source-domain (*Photo*, *Art* and *Sketch*) representations are colored by class and domain. Target-domain (*Cartoon*) representations are colored by class. Refer to the analysis of the domain-linked $\mathcal{Y}_L$ class generalization (solid circle) and domain-invariant learning evidence (broken circle) found in Sec. 6.2.

