# OpenReview forum: "Domain Generalization for Domain-Linked Classes"
_ICLR.cc/2024/Conference — Submitted to ICLR 2024_

### Official Review · Reviewer_UVpZ · 2023-10-20

**Soundness:** 2 fair
**Presentation:** 3 good
**Contribution:** 2 fair
**Rating:** 3
**Confidence:** 5

**Summary:**

This paper delves into the realm of Domain Generalization (DG), emphasizing the challenges posed by domain-linked classes, which are specific to certain domains and thus present significant hurdles in generalization. The authors introduce an algorithm, Fair and cONtrastive feature-space regularization algorithm for Domain-linked DG (FOND), designed to enhance the generalizability of domain-linked classes by leveraging representations from domain-shared classes. Through extensive experiments, FOND purportedly demonstrates state-of-the-art performance in DG tasks for domain-linked classes, provided a sufficient number of domain-shared classes are available. The paper also offers theoretical insights into the factors influencing the performance of domain-linked classes.

**Strengths:**

Novelty: The paper addresses a less-explored area in DG — the challenge posed by domain-linked classes, which significantly hinders the performance of generalization models.

Quality: The introduction of the FOND algorithm, which aims to improve the generalizability of domain-linked classes by utilizing domain-shared class representations, is a noteworthy methodological contribution.

**Weaknesses:**

1. Significance: The practical applicability of the research is questionable, given that the empirical validation is conducted on synthetic datasets, which may not effectively simulate real-world complexities.

2. Quality: The theoretical analysis lacks depth, presenting generalized bounds without significant divergence from existing domain generalization theories, thereby offering limited novel insights.

3. Novelty: The paper's innovation is constrained, primarily adapting existing fairness methods to a new context. The complexity introduced in the loss function isn't justified adequately.

4. Clarity: The paper could benefit from a more coherent presentation of ideas, especially concerning the algorithm's design and the theoretical underpinnings.

**Questions:**

1. Could you elaborate on the choice of synthetic datasets for validation? How do these datasets simulate the challenges of real-world applications?

2. The theoretical analysis seems to align closely with established domain generalization theories. Could you elucidate the novel contributions of your theoretical insights?

3. The FOND algorithm introduces considerable complexity, especially in the loss function. Can you justify this complexity in relation to the performance gains observed?

4. How does the FOND algorithm ensure the transfer of useful representations between domain-shared and domain-linked classes? Is there a mechanism to prevent the transfer of domain-specific biases?

5. Given the focus on domain-linked classes, could the proposed method be adapted to scenarios with fewer or no domain-shared classes?

---

> ### Author Response · Authors · 2023-11-16
>
> We thank you for your review. Below we will address the comments.
>
> **1. Misconception about the evaluation datasets:**
>
> > W1 - Significance: The practical applicability of the research is questionable, given that the empirical validation is conducted on synthetic datasets...
> > Q1 - Could you elaborate...
>
> The reviewer is mistaken. The dataset used – PACS, VLCS and OfficeHome are not synthetic and incorporate real-world images as shown in Fig. 2. Moreover, all contemporary domain generalization works use these dataset for evaluation, and therefore our work cannot be evaluated against an unrealistic benchmark.
>
> **2. Significance of the theoretical result:**
> > W2 - Quality: The theoretical analysis lacks depth...
> > Q2 - The theoretical analysis seems to align closely with established domain generalization theories....
>
> We would like to respectfully disagree. Our main theoretical result is the first result in DG literature which illuminates domain-linked DG challenges; this is extremely important in light of recent calls to develop real world DG techniques. Moreover, existing results also assume that all classes are represented in all domains; this does not reflect real-world challenges.
>
> Furthermore, while existing results are mainly focused on theoretical analysis only, our work leverages the analysis to motivate the proposed approach, whose impacts are further corroborated by in the empirical evaluation where we see that higher number of shared classes helps learning for domain-linked learning. Therefore, while our analysis, like other DG results, uses well-known results in the literature [1,2,3], it is extremely non-trivial and establishes the conditions under which domain-shared classes are better-off than domain linked ones.
>
> **3. Comments about algorithm complexity:**
>
> > Q3 - The FOND algorithm introduces considerable complexity, especially in the loss function. Can you justify this complexity in relation to the performance gains observed?
> > Q4 - How does the FOND algorithm ensure the transfer of useful representations between domain-shared and domain-linked classes? Is there a mechanism to prevent the transfer of domain-specific biases?
>
> 1. From our theoretical and experimental results we observe that domain-linked classes are inherently disadvantaged. Therefore, our aim is to transfer useful representations from domain-shared ones.
> 2. To prevent the transfer of domain-specific biases FOND imposes that inter-domain positive and intra-domain negative comparisons should be amplified via the $\beta$ and $\alpha$ functions respectively. Even in the absence of fairness we observed that the inclusion of both these parameters (FOND\F) yields a **+1.5%** performance improvement.
> 3. To ensure the transfer of useful representations, the FOND objective enforces that learned representations must benefit both domain-shared and domain-linked classes via the fairness objective. This, to effect, encourages the model to ignore domain-specific representations for domain-linked classes and identify generalizable domain-invariant representations from domain-shared classes. Consequently, we observe a **+10%** performance improvement averaged across all datasets and shared-class settings.
> 4. As a result FONDS learning objectives are aligned with our theoretical insights, and are required to accomplish domain-linked generalization.
>
> **4. The case of fewer or no domain-shared classes**
> > Q5 Given the focus on...
>
> This question motivated our decision to create "low" and "high" shared settings. Our methodology shows how domain-shared classes can be used to boost domain-linked performance and our future work involves settings where domain-shared classes cannot be observed (or added) at all.
>
> **5. Comments about clarity and novelty (regarding fairness):**
>
> The paper contributes on multiple fronts, one of which is its novel use of fairness for domain-linked classes (see Sec. 1 and 3 for other contributions and details). Further motivations for the proposed approach are developed in Sec. 4, followed by the methodology in Sec. 5.
>
> Conventional fairness methods reduce distance/divergence metrics between the model’s output on classes across protected attributes [1]; note this requires a class to be expressed by different protected attributes. However, fairness in our context cannot be applied to domains since some classes cannot be observed in multiple domains; this makes the domain-linked setting highly non-trivial.
>
> This is the first work to propose a way to impose fairness in the context of DG aiming to draw useful discriminative representations for domain-linked classes.
>
> **References:**
>
> [1] Eastwood, Cian, et al. "Probable domain generalization via quantile risk minimization." NeurIPS (2022)
>
> [2] Zhang, Guojun, et al. "Quantifying and improving transferability in domain generalization." NeurIPS (2021)
>
> [3] Robey, Alexander, George J. Pappas, and Hamed Hassani. "Model-based domain generalization." NeurIPS (2021)

---

> > ### Comment · Reviewer_UVpZ · 2023-11-21
> >
> > As for W1, my primary concern is that this paper only rebuilds the standard DG datasets rather than proposes or finds a real-world application that has a suitable dataset for the new setting. If there is no fitting dataset existing in the real world, how can you say that the new setting is practical and significant?
> >
> > As for W2, I cannot get the point what are the theoretical contributions of this paper. In my opinion, modifying the existing theory to the new setting is not novel.
> >
> > Based on the response, unfortunately, I have to lower the score.

---

> > > ### Author Response · Authors · 2023-11-21
> > >
> > > **Real World Complexities**
> > >
> > > We respectfully disagree that our datasets do not validate our proposed methods performance in real-world settings. Our validation datasets were specifically selected to simulate four real-world complexities:
> > > 1. Data Scarcity: PACS, VLCS and OfficeHome are the most popular small scale datasets in DG literature [1,2], i.e. ~10,000 images and only 4 domains.
> > > 2. Domain-variances: When aggregating sources in the real-world there are variances in…
> > > - ...class-discriminative characteristics between domains. For example, in PACS, while texture and colour is a powerful descriptor in the photo domain, it is unavailable in the sketch domain.
> > > - ...the environments/contexts of objects between domains. In OfficeHome, the same classes are represented in real-world settings or without backgrounds
> > > - ...scaling, lighting and perspectives of objects between domains such as in VLCS
> > > 3. Task-size: We also examine challenges that may arise when the number of classifications required is moderate, i.e. 5 or 7, and very large, i.e. 65.
> > > 4. Class-distribution: By creating "high" and "low" settings for each dataset we examine different degrees of domain-shared$ness$ in the real-world
> > >
> > >
> > > **Theoretical Contributions**
> > >
> > > Our theoretical contributions are to establish the conditions under which domain-shared classes are better off than domain linked ones. This is critical in guiding our proposed methodology. This work is the first to provide this analysis, experimentally demonstrate the insufficiencies of SOTA DG methods and proposed a solution that outperforms all.

---

### Official Review · Reviewer_8VB1 · 2023-10-30

**Soundness:** 2 fair
**Presentation:** 2 fair
**Contribution:** 2 fair
**Rating:** 5
**Confidence:** 4

**Summary:**

This paper presents a novel task of domain generalization and devises an algorithm aimed at acquiring generalizable representations for domain-linked classes by transferring valuable insights from domain-shared classes.

**Strengths:**

- This paper introduces a new setting of domain generalization where classes can be domain-shared or domain-linked.
- The proposed method applies fairness for the domain-linked classes.

**Weaknesses:**

(1) In section 5.2, the description of fairness is somewhat unclear. Is $M$ referring to the model, specifically the neural network? If so, it seems that the fairness loss is intended to reduce the classification loss gap between domain-linked and domain-shared classes, suggesting that minimizing the fairness loss aims to make the classification loss for both types of classes have similar values during training. However, it would be helpful to clarify how exactly this loss relates to fairness.

(2) Does $\beta$ in equation (4) have different values for each domain? If $\beta$ is a unique value for all domains, then equation (4) can be rewritten as $...log\frac{\alpha}{\beta} \frac{exp(...)}{\sum exp(...)}$. In this case, should we use $\frac{\alpha}{\beta}$ as one hyperparameter instead of two separate hyperparameters ($\alpha$ and $\beta$)? If so, $\frac{\alpha}{\beta}$ would be similar to $\lambda_{xdom}$.

(3) In section A.3 of the appendix, the hyper-parameter selection and model selection process is not quite clear. It references evaluation settings for domains without distinguishing the source and target domains. Does the selection process use the target domain to evaluate the performance?

**Questions:**

Please refer to the Weakness.

---

> ### Author Response · Authors · 2023-11-16
>
> We thank the reviewer for their clarifying questions. We provide details below. All results are reproducible, and further details are in supplementary and the code.
>
> **1. A note about the questions/weaknesses:**
>
> It seems that the Reviewer is enthusiastic about the work, and we are happy to ensure that the reviewer is satisfied with the clarity to improve the presentation. But since none of their questions relate to the quality of contribution, technical novelty, correctness, or experimentation quality, we will strive to address them in the discussion period and we sincerely hope that the Reviewer can reconsider their scores.
>
> **2. Fairness objective clarification:**
>
> > W1 - In section 5.2, the description of fairness is somewhat unclear. Is M referring to the model, specifically the neural network? If so, it seems that the fairness loss is intended to reduce the classification loss gap between domain-linked and domain-shared classes, suggesting that minimizing the fairness loss aims to make the classification loss for both types of classes have similar values during training. However, it would be helpful to clarify how exactly this loss relates to fairness.
>
> Indeed M refers to the model and conventional fairness objectives aim to make the model's performance independent of protected attributes [1-5] (e.g. gender, race, geography). This is formulated as learning objectives that minimize distance/divergence metrics between the model’s output on a certain class with different protected attributes [1]; note this requires a class to be expressed in different protected attributes. However, the reviewer is correct that fairness in our context cannot be applied to domains since some classes cannot be observed in multiple domains. Therefore, we define a fairness objective that seeks to promote representations that benefit both domain-shared and domain-linked classes. We observe that this regularization yields a +10% domain-linked accuracy performance improvement when tested on novel domains.
>
> **3. Clarifying $\beta$:**
> > W2 - Does [beta] in equation (4)...
>
> $\beta$ does not have different values for each domain. $\beta$ is a function, $\beta(z_i,z_j)$, that outputs a value (a hyper-parameter $>1$ ) when a pair of samples $z_i,z_j$ are from different classes (negative) but the same domain (intra-domain). Therefore the $\beta$ function (outputs either 1 or b) cannot be factored out of the summation $\sum\limits \beta \exp(...)$ as you would a constant.
>
> **4. Clarifying hyper-parameter selection:**
> > W3 - In section A.3 of the appendix, the hyper-parameter selection and model selection process is not quite clear. It references evaluation settings for domains without distinguishing the source and target domains. Does the selection process use the target domain to evaluate the performance?
>
> - Model Selection Data: 20% of each source (training) domain held-out during training
> - Model Evaluation Data: 100% of the target (testing) domain
>
> The target domain data must not be used for model/hyper-parameter selection since this data is to remain "unseen" and emulate real-world conditions [6].
>
> **References:**
>
> [1] Wang, Zeyu, et al. "Towards fairness in visual recognition: Effective strategies for bias mitigation." Proceedings of the IEEE/CVF conference on computer vision and pattern recognition. 2020.
>
> [2] Pham, Thai-Hoang, Xueru Zhang, and Ping Zhang. "Fairness and accuracy under domain generalization." International Conference on Learning Representations (ICLR). 2023.
>
> [3] Makhlouf, Karima, Sami Zhioua, and Catuscia Palamidessi. "Machine learning fairness notions: Bridging the gap with real-world applications." Information Processing & Management 58.5 (2021): 102642.
>
> [4] Dwork, Cynthia, et al. "Fairness through awareness." Proceedings of the 3rd Innovations in Theoretical Computer Science Conference on-ITCS'12. ACM Press, 2012.
>
> [5] Madras, David, et al. "Learning adversarially fair and transferable representations." International Conference on Machine Learning. PMLR, 2018.
>
> [6] Gulrajani, Ishaan, and David Lopez-Paz. "In Search of Lost Domain Generalization." International Conference on Learning Representations. 2020.

---

> > ### Comment · Reviewer_8VB1 · 2023-11-22
> >
> > Thank you for your response. Despite the author's clarification of certain details, the primary concern persists regarding the explanation of fairness and the corresponding loss. Besides, applying fairness or causality to domain generalization is not quite novel [1,2].
> >
> > [1] Mahajan, Divyat, Shruti Tople, and Amit Sharma. Domain generalization using causal matching. ICML, 2021.
> >
> > [2] Lv, Fangrui, et al. Causality inspired representation learning for domain generalization. CVPR. 2022.

---

### Official Review · Reviewer_x5U3 · 2023-11-01

**Soundness:** 3 good
**Presentation:** 3 good
**Contribution:** 2 fair
**Rating:** 5
**Confidence:** 4

**Summary:**

This paper addresses a key challenge in Domain Generalization (DG): the difficulty in generalizing to unseen target domains when classes are unique to specific domains (domain-linked). The authors introduce the concept of domain-linked classes in DG and propose the FOND algorithm, which enhances generalization by leveraging knowledge from domain-shared classes. Through comprehensive experiments, they demonstrate that FOND achieves state-of-the-art results in DG tasks, particularly for domain-linked classes. The paper also offers theoretical and practical insights into managing domain-linked class generalizability in real-world scenarios.

**Strengths:**

Indeed, modeling that explicitly considers the relationship between domains and classes is not extensively developed in existing methodologies. In this regard, addressing this specific aspect presents a novel approach to problem-solving in the field. This innovative focus could provide significant advancements in understanding and tackling domain-specific challenges.

It's reasonable to assume that domain-linked classes might have limited data compared to domain-shared classes. If the information from the more abundant domain-shared class data can be effectively utilized for the learning of domain-linked classes, it could indeed be beneficial. This approach seems quite plausible and potentially impactful in addressing data scarcity challenges in specific domains.

**Weaknesses:**

The simplicity of the proposed methodology, which essentially relies on contrastive learning based on domain-shared classes and aligns the losses between domain-linked and domain-shared classes, does seem straightforward. While leveraging information from domain-shared classes to inform domain-linked classes could be beneficial, it's understandable to question whether such loss matching alone suffices to supply rich information.

Furthermore, the connection between merely aligning loss magnitudes and achieving fairness metrics seems tenuous. A deeper, more nuanced approach might be necessary to ensure that the model not only aligns superficial loss values but also genuinely captures and transfers the underlying complexities and variances of the classes across different domains.

The term "domain-linked class," used to describe classes that correspond one-to-one with a specific domain, does not seem particularly intuitive. Just recommend utilizing an other word.

The assumption of awareness on domain-shared classes and domain-linked classes are also not realistic.

The likelihood of encountering domain-linked classes in real-world problems may not be immediately apparent or intuitive. Can you provide a clear, real-world example where such classes prominently emerge?

The result of Theorem 1 appears overly direct. Its derivation through the PAC-Bayes bound seems far too straightforward, making it questionable to regard this as a true theorem.

**Questions:**

Q1. Is it common in this field to define a dataset comprising both inputs and labels as a domain, as done in this paper?

Q2. (Same as weaknesses) The likelihood of encountering domain-linked classes in real-world problems may not be immediately apparent or intuitive. Can you provide a clear, real-world example where such classes prominently emerge?

---

> ### Author Response · Authors · 2023-11-16
>
> We thank you for your review. We address the main comments below.
>
> **1. A note about the misconception in the main concern:**
>
> It seems that the scores are based on the misconception that the availability of labels and domain membership of classes is unrealistic; domain generalization works assume that this information is available [1]. Moreover, we do not assume anything about the target, which may contain any subset of classes observed in the source domain. Therefore, we sincerely urge the Reviewer to reconsider their scores.
>
> **2. Comments about the algorithm structure:**
> > W1 - The simplicity of the proposed methodology ...
>
> We would like to respectfully disagree. Drawing from contrastive and fairness learning literature to address domain-linked class DG is highly non-trivial. Fairness objectives minimise divergences between a model’s output on a class regardless of protected attribute [4]; this requires a class to be expressed in different protected attributes. However, we cannot compare model performance for a class across domains since some are domain-linked. To overcome these challenges we use loss values across domain-shared and domain-linked classes (eq 5) as a surrogate for fairness. Although counter-intuitive, successful DG approaches do not seek to capture the variances across different domains but to be **invariant** to them; see [1] and the references therein. Therefore, our method draws discriminative representations from domain-shared classes to benefit domain-linked ones.
>
> > W2 - ...more nuanced approach...genuinely captures...underlying complexities...
>
> We also respectfully disagree that FOND does not genuinely capture and transfer the underlying complexities. On the contrary, our results and ablations consistently demonstrate how FOND significantly outperforms, even when tested on novel domains, in our rigorous experimental set-up. Moreover, the T-SNE plots demonstrate how our methodology is able to transfer discriminative characteristics for domain-linked classes. We also introduce this task to the community to spark further research, an additional goal of our work.
>
> **3. Misconception about domain/class label accessibility in Domain Generalization:**
> > Q1 - Is it common in this field to define a dataset comprising both inputs and labels as a domain, as done in this paper?
>
> > W4 - The assumption of awareness on domain-shared classes and domain-linked classes are also not realistic.
>
> Yes, datasets such as PACS, VLCS, and OfficeHome are widely used as different domains in domain generalization literature [1,2,3]. Since DG assumes that source domain identities and labels are given, the assumption is in complete alignment with the literature.
>
> **4. Domain-Linked classes in the real-world:**
> > Q2 - The likelihood of encountering domain-linked classes in real-world...
>
> Domain-linked classes occur in real-world applications such as defect detection for quality control applications in manufacturing and food inspection.  In these applications, defect types (classes) vary by products (domain), with certain defects being observed only in certain products. This makes defect detection in new future products extremely challenging where, due to unobserved defects in new contexts, the accuracies of defect detection vary by their type. Specifically, since defects are linked to products, their detection in new products suffers since the model learns spurious correlations between the defect and the product. With its focus on overall accuracy, the current domain generalization methods are inadequate. This is where domain-linked domain generalization can greatly alleviate the performance gap by addressing generalization under these data scarcity issues.
>
> **5. Comments regarding nomenclature:**
> > The result of Theorem 1 appears overly direct...
>
> We use the term “Theorem” as opposed to “Proposition” since the result encapsulates an important result of the paper. This is the first result in domain generalization literature which establishes the conditions under which domain-shared classes are better off than domain linked ones. We are open to renaming based on reviewer’s comments, but we would like to respectfully disagree that this is a weakness of the work since naming can be fixed.
>
> > The term "domain-linked class," ... does not seem particularly intuitive...
>
> Thank you for the feedback. We also considered 'domain-specific', but this term is also used in the literature in other contexts. Since this task has never been considered, we chose “domain-linked”,  but we are open to other recommendations.
>
> **References:**
>
> [1] Zhou et al. Domain generalization: A survey. TPAMI 2022.
>
> [2] Gulrajani, Ishaan, and David Lopez-Paz. "In Search of Lost Domain Generalization." ICLR. 2020.
>
> [3] Zhang, Xingxuan, et al. "Nico++: Towards better benchmarking for domain generalization." CVPR. 2023.
>
> [4] Wang, Zeyu, et al. "Towards fairness in visual recognition: Effective strategies for bias mitigation." CVPR. 2020.

---

> > ### Comment · Reviewer_x5U3 · 2023-11-22
> > **Response on the author's rebuttal**
> >
> > Most concerns are resolved by the author's response.
> > I would like to increase my score from 3 to 5.
> > The novelty issues remain unchanged.

---

### Meta-Review · Area_Chair_65mR · 2023-12-06

**Metareview:**

This paper studies an interesting setting of domain generalization by emphasizing the challenges by domain-linked classes. The paper is well organized and clearly written, and the proposed method is plausible. Reviewers provided comprehensive comments on assumptions, notations, technical details, experiments, etc. However, the responses from authors failed to address all of these concerns. During the discussion period, the reviewers and ACs agreed that the novelty of this work is below the bar of ICLR.

**Justification For Why Not Higher Score:**

The novelty of this work is below the bar of ICLR. Also, some comments from reviewers (e.g., explanation of fairness and the corresponding loss) were not well addressed in the rebuttal.

**Justification For Why Not Lower Score:**

N/A

---

### Decision · Program_Chairs · 2024-01-16

Reject